# Possibility for Prevention of Type 2 Diabetes Mellitus and Dementia Using Three Kinds of Brown Rice Blends after High-Pressure Treatment

**DOI:** 10.3390/foods11060818

**Published:** 2022-03-12

**Authors:** Sumiko Nakamura, Takeshi Ikeuchi, Aki Araki, Kensaku Kasuga, Kenichi Watanabe, Masao Hirayama, Mitsutoshi Ito, Ken’ichi Ohtsubo

**Affiliations:** 1Faculty of Applied Life Sciences, Niigata University of Pharmacy and Applied Life Sciences, 265-1, Higashijima, Akiha-ku, Niigata 956-8603, Japan; snaka@nupals.ac.jp (S.N.); hirayama@nbrp.co.jp (M.H.); m-ito@nupals.ac.jp (M.I.); 2Brain Research Institute, Niigata University, 1-757, Asahimachidori, Chuo-ku, Niigata 951-8585, Japan; araki-aki@bri.niigata-u.ac.jp (A.A.); ken39@bri.niigata-u.ac.jp (K.K.); 3Faculty of Medicine, Niigata University, 1-757, Asahimachidori, Chuo-ku, Niigata 951-8585, Japan; wataken@med.niigata-u.ac.jp

**Keywords:** wax-free brown rice, waxy black rice bran, n-6/n-3 ratio, RS, insoluble dietary fiber, antioxidative activity, free ferulic acid, anthocyanin

## Abstract

As it has been reported that type 2 diabetes mellitus increases the risk of Alzheimer’s disease, we investigated how to prevent type 2 diabetes and dementia using biofunctional boiled rice. We adopted unpolished super-hard rice (SHBR) for diabetes and wax-free unpolished black rice (WFBBR) for dementia and blended those with ordinary non-polished rice (KBR) (blending ratio 4:4:2), adding 2.5% waxy black rice bran (WBB) and 0.3% rice oil after high-pressure treatment (HPT) (WFBSK) to improve its palatability. This boiled rice is rich in dietary fiber, anthocyanin, free ferulic acid and β-secretase inhibitory activity. A randomized, parallel-group comparison study was conducted for 12 weeks with 24 subjects, using Cognitrax to evaluate their cognitive function primarily. Furthermore, as the secondary purpose, we performed a single-dose test for postprandial blood glucose and insulin secretion at the end of the human intervention test. After 12 weeks, consumers of the WFBSK rice exhibited significant improvement in language memory by cognitive test battery compared with those who consumed the control white rice (*p* < 0.05). Moreover, subjects who consumed the WFBSK rice had lower insulin secretion levels than those who consumed the control polished rice (*p* < 0.05).

## 1. Introduction

According to a report by the IDF (International Diabetes Federation), about 537 million people were candidates or patients with diabetes in 2021 [1] and the number of patients with dementia was about 46 million in the world in 2015 [2].

Dementia is a syndrome where there is deterioration in cognitive function that leads to impairment of the activities of daily living (ADL). Dementia may be caused by a variety of diseases and injuries that primarily or secondarily impair the brain function. Among the dementia patients, most have Alzheimer’s disease (AD); 60–70% of the patients with dementia suffer from AD [3]. Nowadays, dementia is ranked as the seventh main cause of death among all diseases. Furthermore, it tends to lead to the disability and dependency of aged people all over the world [3]. Cognitive function consists of multiple domains such as memory, attention, language and executive function, all of which are essential for our daily life [4]. Cognitive disabilities caused by dementia negatively affect the quality of life (QOL) of elderly patients [5].

It is well-recognized that the prevalence of dementia is higher in diabetic patients than in non-diabetic subjects [6].

There are many scientific reports about the close relationship between insulin resistance and lifestyle diseases, such as type 2 diabetes and dementia [7,8,9,10,11,12,13,14]. As patients with diabetes have been markedly increasing in number worldwide, the development of a suitable method for its treatment or prevention is very important. The World Health Organization (WHO) and Food and Agriculture Organization of the United Nations (FAO) recommend foods with a low glycemic index (GI) to prevent diabetes [3]. The concept of GI was introduced by Jenkins et al. [15], and low GI and glycemic load diets have become popular to prevent chronic diseases, such as cardiovascular disease, diabetes, cancer and obesity [16], although they all have limited efficacy.

Early intervention for maintaining cognitive function is one of the important factors for successful aging [17]. To investigate dementia patients, the human intervention test is indispensable. Dubois et al. [18] described multi-component interventions targeting several risk factors simultaneously, such as “FINGER”, might be needed for optimal preventative effects, to improve and maintain cognitive functioning.

Many researchers reported that the progression of cognitive decline may be considerably affected by various lifestyle factors, such as foods and drinks [4]. For example, it was reported that cognitive decline was prevented by food components such as minerals, polyphenols, flavonoids, vitamins, omega-3 PUFAs, etc. [19].

Findings from prospective studies suggest that greater adherence to the Mediterranean diet may be associated with slower cognitive decline and reduced risk of AD [20,21]. Core et al. [22]. showed that reduced levels or intake of omega-3 fatty acids or fish are associated with increased risk of age-related cognitive decline or dementia such as AD.

Rice (*Oryza sativa* L.) is one of the most important crops, cultivated in over 100 countries around the world, and is a staple food for about half of the world population [23]. As rice consumers, especially Asian people, eat it almost every day, it is very promising that its biofunctionality, such as prevention of diabetes and dementia, would be very effective. Unpolished rice grains contain more nutritional components, such as dietary fibers, phytic acids, flavonoids, tocopherols, γ-oryzanol and E and B vitamins, than ordinary milled rice grains [24]. The germ and bran layers (about 10% of a rice kernel) removed during the milling procedure are rich in proteins, lipids (γ-oryzanol, ferulic acid, sterol, wax, ceramide, phytin and inositol), fiber, minerals, tocotrienols, tocopherols and B-complex vitamins (B1 and B6) [24]. The use of rice bran as food and feed from 1970 to 1998 was recently reviewed, concerning the control of chronic degenerative diseases such as hypercholesterolemia [25]. Juliano showed that stabilized rice bran is mildly crunchy and slightly sweet, and has a mild toasted aroma [25].

The glycemic effect of food depends on numerous factors such as the microstructure of starch, amylose content and amylopectin chain-length distributions [26,27]. In a previous study, we developed a novel method for inhibiting postprandial blood glucose levels in Sprague–Dawley rats by preparing cooked rice grains from amylose extender (*ae*) mutants of rice soaked with functional food ingredients [28,29]. Several studies have reported the development of highly resistant starch rice as well as high-amylose and high-dietary fiber rice via physical or chemical mutation [30,31]. The cooked grains of super-hard rice cultivars are hard and non-sticky because they lack starch branching enzyme IIb and contain many super-long chains (SLCs) [29]. They are promising in terms of their biofunctionality, such as diabetes prevention and reduction of obesity [32]. Pigmented rice contains polyphenol substances, such as anthocyanins and tannins. Black or purple rice gets its color from anthocyanin pigments, which are known to have free-radical scavenging and antioxidant capacities [33]. Black rice may have antiatherogenic activity and may improve certain metabolic pathways associated with diets high in fructose [34,35]. Red rice is known to be rich in minerals, such as iron and zinc, while black and purple rice are especially high in protein, fat and crude fiber [34].

The cooked grains of unpolished rice are too hard and non-sticky for the consumers. For that reason, Watanabe et al. [36] showed that wax-free brown rice (WFBR), which is unpolished rice with only the wax layer removed, keeping other nutrients in the bran layer, is promising for a low-GI and high dietary fibers and vitamins, and its taste is improved. Furthermore, high-pressure treatment (HPT) is very useful in the food industry [37]. The merits of the high pressure are to avoid the destruction of covalent bonding and to keep the natural flavor, taste, and nutrients. HPT is the technological process that has the least effect on heat-labile water-soluble vitamins, thus contributing to the preservation of the nutritional quality of foodstuffs [38]. HPT was reported to be useful for improving the texture of cooked rice without denaturation of enzyme activities, and it led to an increase of free amino acids and change in enzyme activities [39].

The objective of this study was to develop multifunctional boiled rice, which would be useful for preventing type 2 diabetes and dementia, using black rice and super-hard rice with the aid of HPT and wax-free treatment. The antioxidative capacity of black rice and low-GI of super-hard rice would contribute to preventing the onset of diabetes and dementia.

## 2. Materials and Methods

### 2.1. Materials

The brown rice of super-hard rice cultivar Niigata 129 go (not registered), the high-quality rice Koshihikari; (registration number in Japan; 8539), waxy black rice Shiho (registration number in Japan; 11846) and black rice Okunomurasaki (registration number in Japan; 11088), were cultivated by the Niigata Prefecture Agriculture Research Institute in 2020, and red rice (Beniroman) and waxy red rice (Yuyake mochi) were purchased from a local market. The wax-free brown rice was manufactured by Toyo Rice (Co., Ltd., Wakayama, Japan) in 2020.

### 2.2. Preparation of Waxy Black Rice Bran

The bran of waxy black rice (Shihou cultivar) was prepared by polishing using an experimental friction-type rice milling machine (Yamamoto Seisakusyo Co., Tendo, Japan) to a milling yield of 90%.

### 2.3. Food Processing of Experimental Meal

WFBSK rice (WFBBR:SHBR:KBR = 4:4:2) was combined with 2.5% (*w*/*w*) waxy black rice bran (WBB) and 0.3% (*w*/*w*) rice oil (Tsuno Food Industrial Co., Ltd., Wakayama, Japan), obtained by treating these at 200 MPa for 2 min in a high-pressure machine (Ishikawajima-Harima Heavy Industries Co., Ltd., Tokyo, Japan), and cooked rice was prepared by Echigo Seika, Co., Ltd. (Nagaoka, Japan). Commercial aseptic cooked rice (using polished rice of high-quality rice cultivar, Koshihikari, as material) was prepared with the same condition with the abovementioned procedure, by Echigo Seika, Co., Ltd., and subjected to the analyses, as well as being used as a control meal in the human intervention test.

### 2.4. Measurement of Moisture Content of Rice Flour

The moisture content of the brown rice and cooked brown rice were measured using an oven-dry method. Accordingly, 2 g of brown rice sample was dried for 1 h at 135 °C, while 2 g cooked rice samples were dried for 3 h at 135 °C.

### 2.5. Measurement of Water Absorption Rate (WAR) of Wax-Free Brown Rice (WFBR), Brown Rice and White Rice Grains

Each sample grain (5.0 g) was socked in 50 mL distilled water at 25 °C for 15 min, 30 min, 60 min, 90 min, 120 min or 180 min. After draining, the clear supernatant was removed, and we weighed the sample grains (Wa). The water absorption rate (WAR) was calculated as WAR (%) = (Wa − 5)/5 × 100.

### 2.6. Textural Properties of Cooked Rice

The physical properties of cooked rice grains were measured based on bulk measurement (10 g), using a My Boy System Tensipresser (Taketomo Electric Co., Tokyo, Japan) according to the method described by Okadome et al. [40]. For standard samples, milled rice (10 g) was combined with 14 g distilled water (coefficient [gross water volume/dry matter weight]: 1.77, calculated for each sample) in an aluminum cup. After soaking for 1 h, the samples were cooked. The cooked rice samples were kept in the vessel at 25 °C for 2 h and subjected to the measurements. The bulk measurements were repeated five times, and the mean value was calculated.

### 2.7. Measurement of the Fatty Acid Composition of Rice Bran

Measurement of the fatty acid composition of rice bran was carried out by the Food Analysis Technology Center (using a gas chromatography method). A rice bran sample (0.2 g) was extracted with 2 mL hexane and mixed well. After that, 2 M potassium hydroxide−methanol solution (0.2 mL) was added and mixed.

### 2.8. Measurement of RS (Resistant Starch) of Cooked Rice

The resistant starch (RS) was measured according to the AOAC method (2002.02) using an RS assay kit (Megazyme, Ltd., Wicklow, Ireland) except the enzyme reaction time. Freeze-dried rice flours (0.1 g) were treated with a 0.1 M sodium maleate (4 mL) (pH 6.0) buffer solution with enzymes (pancreatin and amyloglucosidase) at 36 °C for 6 h, and then denatured ethanol (99%) (4 mL) was added and the solution was centrifuged. The precipitation was mixed with denatured ethanol (99%) (6 mL) and the reaction mixture was centrifuged, and the process was then repeated. The supernatant was removed, and this was followed by the addition of 2 M potassium hydroxide (2 mL), stirred in ice water for 20 min, and 1.2 M sodium acetate (pH 3.8) (8 mL) and amyloglucosidase (0.1 mL) were added and stirred at 50 °C for 30 min in a water bath. After centrifugation, the supernatant (0.1 mL) was mixed GOPOD (glucose oxidase-peroxidase-aminoantipyrine reagent) (3 mL) and stirred at 50 °C for 20 min in a water bath. The glucose content was measured using a spectrophotometer at 510 nm.

### 2.9. Measurement of Polyphenol Content of Cooked Rice

The polyphenol content of freeze-dried rice samples was determined using the Folin–Ciocalteu method [41]. Each sample (0.1 g) was extracted with 80% ethanol (4 mL) and then centrifuged. The supernatant (1 mL) was mixed with the same volume of Folin–Ciocalteu solution (1 mL) and incubated for 3 min at room temperature, followed by the addition sodium carbonate (5 mL) and incubation at 50 °C for 5 min. Finally, the sample solution was cooled in ice water for 1 h and then centrifuged. Absorbance was measured at 765 nm. Gallic acid was used for calibration.

### 2.10. Measurement of Hydrophilic and Lipophilic Oxygen Radical Absorbance Capacity (H-ORAC and L-ORAC)

The hydrophilic and lipophilic oxygen radical absorbance capacities of freeze-dried rice samples were measured as described by Prior et al. [42]. Trolox calibration solutions were made to obtain a standard curve [43,44]. For the hydrophilic antioxidant assay, freeze-dried rice flours (0.1 g) were extracted with hexane (10 mL), and the hexane layer was removed. Residual hexane was evaporated using a water bath at 70 °C, and the residue was then extracted with acetone/water/acetic acid (70:29.5:0.5, *v*/*v*/*v*) (10 mL). The solution was sonicated (Ultrasonic cleaner 3510J-MTH, Branson Ultrasonics Co, Ltd., Richmond, VA, USA) at 37 °C for 15 min, and then centrifuged. The supernatant was diluted to a 25-mL total volume. For the lipophilic antioxidant assay, freeze-dried rice flours (0.1 g) were extracted with hexane (1 mL), and then centrifuged. The hexane was evaporated using a water bath at 70 °C, and the dried hexane extract was then dissolved in acetone (250 μL) and diluted with 7% randomly methylated β-cyclodextrin (750 μL) (RMCD; 0.7 g methyl-β-cyclodextrin (Sigma-Aldrich Co. LLC, St. Louis, MO, USA) dissolved in 10 mL 50% acetone). Absorbance values were measured at 485 nm (excitation) and 530 nm (emission) using a fluorescent microplate reader (Grating Based Multimode Reader SH-9000, Corona Electric Co, Ltd., Hitachinaka-shi, Japan).

### 2.11. Measurement of the Ferulic Acid Composition of Cooked Rice

Measurement of the ferulic acid composition of rice bran was carried out by the Japan Food Research Laboratories (using microbiological assays and high-performance liquid chromatography–mass spectrometry).

### 2.12. β-Secretase Inhibitory Activity

The β-secretase (BACE1) inhibitory activity of freeze-dried cooked rice was measured using a BACE1 activity detection kit (Fluorescent; Sigma-Aldrich Co. LLC.). Freeze-dried rice flour (0.1 g) was extracted with 10 mM acetate buffer solution (0.5 mL) (pH 5.0, including 0.1% Triton and 0.05% CHAPS) for 1 h, and then centrifuged. The absorbance values were measured at 320 nm (excitation) and 405 nm (emission) using a fluorescent microplate reader (Grating Based Multimode Reader SH-9000, Corona Electric Co, Ltd.) [45].

### 2.13. Measurement of the Dietary Fiber of Cooked Rice

Measurement of the dietary fiber of cooked rice was carried out by the Japan Food Research Laboratories (using microbiological assays and high-performance liquid chromatography–mass spectrometry).

### 2.14. Measurement of the Anthocyanin Content of Cooked Rice

The anthocyanin content of cooked rice flour was measured based on the pH different method [46]. Freeze-dried rice flours samples (1.0 g) were extracted with 9 mL methanol/deionized water/trifluoroacetic acid (2:3:0.025, *v*/*v*/*v*) at room temperature for 2 min by strong vortexing. The solution was then sonicated (Ultrasonic cleaner 3510J-MTH, Branson Ultrasonics Co, Ltd.) at 37 °C for 5 min and soaked in a water bath at 37 °C for 10 min, then centrifuged for 2 min at 3000× *g.* The supernatant was removed and transferred to a 25-mL volumetric flask. The precipitate was extracted with 8 mL methanol/deionized water/trifluoroacetic acid (2:3:0.025, *v*/*v*/*v*) and the process repeated two times. The supernatant was removed to a volumetric flask and diluted to a 25-mL total volume, and the solution was twofold diluted with pH 1.0 buffer (0.025 M potassium chloride) and another pH 4.5 buffer (0.4 M sodium acetate). The absorbance was measured at 520 nm and 700 nm. The anthocyanin content was expressed as the cyanidin 3-glucoside.

### 2.15. Sensory Evaluation

The sensory test was carried out by the method reported in our previous paper [47]. Seven-grade ranking was used to evaluate the six attributes, such as appearance, aroma, hardness, taste, stickiness, and overall evaluation by 10 trained taste panelists.

### 2.16. Study Design of Human Intervention Test

The study protocol for human test subjects was approved by Niigata Bio-Research Park Inc (Niigata, Japan). and approved by the ethics committee for human tests of the Niigata Bio-Research Park and Niigata University of Pharmacy and Applied Life Sciences, and according to the 2014 guidelines issued by the Ministry of Education, Culture, Sports, Science and Technology and the Ministry of Health, Labor and Welfare, Japan. Informed consent was obtained for experimentation with human subjects, and the test was registered as UMIN000044767.

A randomized, parallel-group comparison study was conducted to examine the effect of WFBSK rice. Inclusion criteria were as follows: (1) subjects aged from 50 to 75 years, (2) not diagnosed as having dementia or diabetes, (3) no consumption of supplements that may affect cognition or blood glucose. We recruited 24 healthy subjects for the study. The effect of WFBSK rice on cognitive performance was the primary endpoint. Secondary endpoints included inhibition of abrupt increase in postprandial blood glucose level and change in plasma amyloid-β (Aβ) 42/40 ratio.

Components of the test and control meals are shown in Appendix A. Participants were randomly assigned to one of two groups. Subjects of each block-consumed the assigned test sample meal (one package of boiled rice) containing 64 g (test meal) or 67 g (control meal) carbohydrate once every day for 12 weeks consecutively. After 12 weeks, test subjects got the single-dose test for BGL measurements, where they consumed the assigned meals within 10 min with frequent mastication (30 times was the guideline) and 200 mL of water.

### 2.17. Evaluation of Cognitive Function

The Mini-Mental State Examination Japanese version (Nihon Bunka Kagakusha, Tokyo, Japan) was evaluated at the baseline. The Cognitrax test (Health Solution, Inc., Tokyo, Japan) was utilized to assess the cognitive change by intervention. Cognitrax consists of a computerized test battery, which evaluates multiple cognitive domains, including composite memory, verbal memory, visual memory, processing speed, psychomotor speed, executive function, reaction time, complex attention, simple attention, cognitive flexibility and motor speed. Cognitrax scores have been standardized according to the results from large populations of subjects aged from 7 to 90 years [48].

### 2.18. Measurement of Plasma Concentrations of Aβ1-42 and Aβ1-40

Plasma concentrations of Aβ1-42 and Aβ1-40 were analyzed using a V-PLEX Aβ Peptide Panel 1 (6 × 10^10^) Kit (Meso Scale Discovery, Rockville, MD, USA) with MESO QuickPlex SQ 120 (Meso Scale Diagnostics, LLC, Rockville, MD, USA) according to the manufacturer’s instructions. The intra-assay and inter-assay coefficients of variation were less than 20% for all assays.

### 2.19. Blood Examination

Blood was drawn (in November 2021) under fasting conditions from each subject for analysis of Aβ40, Aβ42, HDL cholesterol, LDL cholesterol, insulin sensitivity and HbA1c. Plasma concentrations of Aβ1-42 and Aβ1-40 were analyzed using a V-PLEX Aβ Peptide Panel 1 (6 × 10^10^) Kit (Meso Scale Discovery, Rockville, MD, USA) with MESO QuickPlex SQ 120 (Meso Scale Diagnostics, LLC, Rockville, MD, USA) according to the manufacturer’s instructions. The intra-assay and inter-assay coefficients of variation were less than 20% for all assays. BGL and insulin levels were measured at 0, 30, 60, 90 and 120 min after two groups of 12 subjects who had eaten two different test meals.

### 2.20. Statistical Analyses

All results were subjected to t-tests and Dunnett’s test using Excel Statistics (version 6, Microsoft Corporation, Tokyo, Japan) and GraphPad Prism V8.4.3 (GraphPad Software, Inc., San Diego, CA, USA). A value of 0.05 < *p* < 0.10 was considered to show the tendency, and a value of *p* < 0.05 was considered to be statistically significant.

## 3. Results and Discussion

### 3.1. Water Absorption Rate (WAR) of WFBR and Brown Rice

The water absorption rates (WARs) of WFBR and brown rice are shown in Figure 1. The WAR of polished Koshihikari rice is shown as a control. As shown in Figure 1, the WFBRs of Koshihikari, Okunomurasaki (black rice) and Niigata 129 go (super-hard rice) were significantly higher than those of brown rice at every soaking time. Moreover, the WAR of Koshihikari WFBR showed almost the same value as the control polished rice for 1 h soaking. Furthermore, the WAR of Okunomurasaki WFBR was significantly higher than that of control polished rice for 2 h soaking, and that of Niigata 129 go (super-hard rice) showed a similar tendency. Kuwada et al. [49] showed that the WAR of brown rice had almost the same value as polished rice for 17 h soaking, because the wax layer of brown rice interferes with water absorption. In a previous study, we reported that the WAR of brown rice becomes higher due to damage of the cell wall by high-pressure treatment (HPT) [39]. As a result, WFBR could be cooked after 1 h soaking, the same as with white rice.

### 3.2. Textural Properties of Cooked Grains of WFBR and Non-Wax-Free Rice

Hardness (force) and toughness (work) are parameters for rigidness, while adhesiveness (work) and stickiness (force) are indicators of tackiness in the textural measurement.

As shown in Table 1, the hardness of Koshihikari and Niigata 129 go WFBR were significantly lower than those of ordinary brown rice, and toughness showed a similar tendency. Moreover, the stickiness of Koshihikari, Okunomurasaki and Niigata 129 go WFBR were significantly higher than those of ordinary brown rice, and adhesion showed a similar tendency. The cooked rice grains after wax-free treatment were softer and stickier than those of non-wax-free brown rice, while their hardness and toughness were lower. Moreover, the ratio of stickiness and hardness of cooked rice with the wax-free treatment were increased 1.4~1.6 times than with the non-wax-free treatment. We estimate that the reason why WFBR was softer and stickier than non-wax-free brown rice is that the former rice grains absorbed water more rapidly and thoroughly. As a result, wax-free treatment markedly improved the textural properties of cooked brown rice. It is now possible to produce palatable cooked rice grains by wax-free treatment. Furthermore, in terms of the whiteness and brightness of cooked rice, WFBR showed significantly higher values than ordinary brown rice, and on the other hand, in terms of a yellowish color, WFBR showed significantly lower values than brown rice (Appendix A). In previous studies, we reported that high-pressure treatment (HPT) improved the physical properties of brown rice [39]. Therefore, wax-free treatment and HPT improved the textural properties of cooked brown rice so that consumers accept them as the table rice they eat every day.

### 3.3. Fatty Acid Compositions of Six Kinds of Rice Bran

In a previous study, we reported the fatty acid compositions of 30 japonica rice cultivars [50]. We found that a low DP (degree of polymerization) of amylopectin cultivars more easily bonded to linolenic acid (18:3n-3) than linoleic acid (18:2n-6) [50]. As shown in Table 2, the fatty acid compositions of six kinds of rice bran were palmitic acid (16.3–18.2%, mean = 17.3%), oleic acid (39.3–46.0%, mean = 42.7%) and linoleic acid (30.6–36.6%, mean = 34.0%), and the n-6/n-3 ratio was 24.4–40.0 (mean = 29.9). The oleic acid contents of Okunomurasaki (black rice) (46.0%) and Niigata 129 go (super-hard rice) (45.3%) were high; that of Shiho (waxy black rice) (43.9%) was intermediate; those of Koshihikari (41.2%), Yuyake mochi (waxy red rice) (40.5%) and Beniroman (red rice) (39.3%) were low. In contrast, the linoleic acid contents of Koshihikari (36.7%), Beniroman (36.6%) and Yuyake mochi (36.1%) were high, and those of Okunomurasaki (32.0%), Shiho (31.7%) and Niigata 129 go (30.6%) were low. Furthermore, the n-6/n-3 ratios of Okunomurasaki (40.0) and Koshihikari (33.4) were high; those of Niigata 129 go (27.8), Yuyake mochi (27.8) and Beniroman (26.1) were intermediate; that of Shihou (24.4), a waxy black rice cultivar, was very low. Simopoulos [51] showed that increased levels of omega-3 PUFA (a low n-6/n-3 ratio) exert suppressive effects against the pathogenesis of several diseases. Waxy rice lipids tend to be richer in palmitic acid but poorer in oleic, and to a lesser extent, in linoleic acids [52]. Our results showed that the n-6/n-3 ratio was negatively correlated with palmitic acid (r = −0.85, *p* < 0.05) (data not shown). As shown in Appendix A, the n-6/n-3 ratios of super-hard rice, waxy black rice, waxy red rice and red rice were low; therefore, they would be biofunctional rice cultivars. As an example of the effects of fatty acid to prevent disease through the diet, eicosapentaenoic acid (EPA) prevented thrombosis and atherosclerosis [53]. Ikemoto and Naganuma [54] reported that the n-6/n-3 ratio of PUFAs would make a very important dietary index because n-3 fatty acids, such as linolenic acid, lead to the generation of eicosapentaenoic acid and docosahexaenoic acid (DHA) in the body.

### 3.4. Nutrient Intake and Structural Components of Three Kinds of Rice Blends

Shown in Table 3A is a diet of blending super-hard brown rice (SHBR), wax-free black rice (WFBBR) and ordinary brown rice (KBR) (4:4:2) with 2.5% waxy black rice bran (WBB) and 0.3% rice oil added after high-pressure treatment (HPT) (WFBSK). In Table 3B, there is a diet of blending super-hard brown rice (SHBR), black/brown rice (BBR) and ordinary brown rice (KBR) (4:4:2) with 2.5% waxy black rice bran (WBB) and 0.3% rice oil added after high-pressure treatment (HPT) (BSK). In Table 3C, there is a diet of control polished Koshihikari rice. The insoluble dietary fiber (ISDF) of WFBSK (9.5 ± 0.6) (g/100 g) was significantly higher than BSK (7.8 ± 0.2) (g/100 g) and the control diet (0.4 ± 0.4) (g/100 g), whereas the water-soluble dietary fiber (SDF) of BSK (0.8 ± 0.1) (g/100 g) was significantly higher than WFBSK (0.3 ± 0.1) (g/100 g) and the control diet (0.1 ± 0.1) (g/100 g). The protein, lipid and ash contents of WFBSK and BSK showed almost the same values, and those values were significantly higher than the control Koshihikari diet. The effect of the β-glucan and dietary fiber contents on the LDL cholesterol and postprandial blood glucose of the subjects were very similar. Several recent studies, in both hypercholesterolemic and healthy subjects, found that the daily consumption of 5 g of β-glucan significantly lowered the total and LDL cholesterol in serum [55,56,57]. As a result, it was shown that the ISDF and total dietary fiber contents were significantly higher, by 1.1 times in the case of both wax-free processing and HPT (Table 3A) versus the case of only HPT (Table 3B).

### 3.5. Measurement of Biofunctinal Properties of Three Kinds of Cooked Rice

Many dietary compounds have been proposed to be important antioxidants, and although there is credible evidence that vitamins E and C are important antioxidants, the evidence is weaker for carotenoids and related plant pigment [58]. The H-ORAC values of WFBSK (77.5 ± 3.4) (μmol TE/100 g FW) and BSK (78.0 ± 1.3) (μmol TE/100 g FW) were significantly higher than the control diet (0.0 ± 0.0) (μmol TE/100 g FW), and the L-ORAC values trended in similar directions to the H-ORAC values. As a result, the total ORAC values of WFBSK and BSK proved to be higher than the control diet by about 80 times, and similarly, the anthocyanin and polyphenol contents of WFBSK and BSK were significantly higher than the control diet, by about eight times. Pigmented rice contains naturally occurring pigmented flavonoids known as anthocyanins. Positive health effects have been reported for the pigments in the bran layer of rice [50]. In a previous study, we reported that the H-ORAC values and available polyphenol content of black rice were increased by HPT [39].

RS is starch that eludes digestion in the small intestine and may ferment in the large intestine. Several studies have reported that long-term consumption of RS might reduce the fasting cholesterol and triglyceride concentrations [59,60]. Yang et al. reported the starch properties of mutant rice, which is rich in resistant starch [30]. Furthermore, Noro et al. showed that Niigata 129 go (Chou 2418) contains long-chain glucans of amylopectin and that the RS content is higher than in high-amylose rice cultivars [61]. Maeda et al. [62] showed that the RS content of mutant rice became slightly higher than in that untreated by HPT, although their physical properties were improved. As shown in Table 4, the RS contents of WFBSK (5.1 ± 0.1%) and BSK (5.1 ± 0.1%) were significantly higher than with the control diet (0.7 ± 0.0%).

BACE1 is the first protease involved in the process of converting APP to Aβ in the brain. As shown in Table 4, the BACE1 inhibitory activities for WFBSK (9.2 ± 0.3%) and BSK (8.8 ± 0.3%) were significantly higher than for the control diet (0.0%). The BACE1 inhibitory activity of pigmented rice bran is much stronger than the inhibitory activities of ordinary rice bran, as reported in our previous paper [45].

Ferulic acid is used as an antioxidant and antimicrobial agent. It is also recognized that ferulic acid exhibits a preventive effect on discoloration in various food products, and a variety of physiological functions such as suppression of Alzheimer’s disease, prevention of muscular fatigue, improvement of hypertension and antitumor activity in the breast, liver and colon [24]. As shown in Table 4, the free ferulic acid levels of WFBSK (0.60 ± 0.00) (mg/100 g) and BSK (0.57 ± 0.01) (mg/100 g) were significantly higher than for the control diet (0.30 ± 0.00) (mg/100 g). These results were presumed to be due to HPT.

As a result, the biofunctional properties of WFBSK tended to be slightly higher than the BSK ones. Therefore, we selected WFBSK as the test diet for the human intervention test.

### 3.6. Textural Properties of Three Kinds of Cooked Rice

As shown in Table 5, the hardness and adhesion of three kinds of cooked rice showed similar values, but the toughness and stickiness of WFBSK and BSK were significantly higher than those of the control diet. As a result, the textures of the cooked WFBSK and BSK rice became acceptable for consumers, in terms of palatability, by HPT.

The biofunctional and physical properties of WFBSK and BSK showed a similar tendency. Therefore, we evaluated the palatability of blended meals by sensory evaluation. We found the “taste” and “overall evaluation”, in sensory analysis of WFBSK meal, were significantly higher than those of BSK meal at the level of 5%, as shown in Table 6. Furthermore, the “taste” and “overall evaluation” of WFBSK meal were improved by adding 0.3% rice oil according to the method for improving the eating quality of cooked rice by adding rice oil [63].

### 3.7. Human Intervention Test for Cognitive Function

We examined the effects of WFBSK meal on the cognitive function of healthy Japanese adults. Twenty-four subjects were randomly assigned to the WFBSK meal or control meal based on the MMSE score. There was no significant difference in subject age, BMI, MMSE score, fasting blood glucose or HbA1c between the WFBSK and control meal groups (Table 7). One pack of WFBSK or the control of white rice was taken every day for a 12-week intervention period (Appendix A). No adverse effects were observed for 12 weeks and all 24 subjects completed the intervention test in good health (Appendix A). Cognitrax was used as a test battery to assess the cognitive function of the participants. Evaluation was performed before and after 12 weeks of intervention. At the baseline, none of the cognitive domains showed significantly different scores between the WFBSK and control meal. As shown in Table 8 and Figure 2, Cognitrax demonstrated that subjects taking WFSFK meal showed significant improvement in the cognitive domain of language memory after 12 weeks (*p* < 0.05).

As biofunctional components to improve the cognitive ability, astaxanthin and sesamin [64], green tea catechins [65], rosemary extracts [66], chlorogenic acids [4], L-theanine [67], astaxanthin and tocotrienol [68] have been reported. It seems that the main component in our sample meal to improve cognitive ability would be anthocyanin because many papers reported the important role of its anti-oxidative capacity, the same as with propolis [69,70], astaxanthin [67,68], green tea catechins [65] and chlorogenic acids [4]. In the present paper, we showed that the WFSFK meal contained a high amount of anthocyanin, a kind of polyphenol.

Although other components, such as gamma-oryzanol, ferulic acid, and tocopherol, in the brown rice and rice oil, could be the candidates for the improvement of cognitive ability, we consider that anthocyanin would be most important because of the stronger ORAC values in the black/brown rice, as shown in Table 4. We will continue our investigation to clarify this it in the next report.

It seems that our results are one of the first examples of effects shown for a foodstuff itself like cooked rice, not for a specified functional component or its extracts.

### 3.8. Changes in Plasma Aβ42/40 Ratio

Blood samples were taken from the 24 subjects after meals before the diets began (baseline) and after 12 weeks. Plasma Aβ40 and Aβ42 were measured, and Aβ42/40 ratio was evaluated as a secondary endpoint. A change in Aβ42/40 ratio from the baseline to the end of the test (after 12 weeks) did not show a significant difference between the subjects who consumed the WFBSK and those who consumed the control meal (Figure 3).

Aβ spices are important components of plaque in the AD brain. The Aβ42/40 ratios in plasma may decline early in the course of AD [71]. High plasma concentrations of Aβ40, especially when combined with low concentrations of Aβ42, indicate an increased risk of dementia [71,72]. The reason why the Aβ 42/40 ratios did not change with the WFBSK meal may be explained by the relatively short period of intervention in this study.

### 3.9. Results of Postprandial Blood Glucose and Insulin Secretion

Panlasigui and Thompson [73] showed that unpolished rice is healthier and more beneficial than polished rice for diabetics and hyperglycemic individuals, and that the glycemic area and GI were 35.4% lower in unpolished rice than in polished rice.

Jung et al. [74] showed that ferulic acid might be beneficial in the treatment of type 2 diabetes because it regulates blood glucose levels (BGLs) by elevating glucokinase activity and producing glycogen in the liver.

The BGL of test subjects at 90 min and 120 min after ingesting the WFBSK meal showed no significantly lower tendency after 12 weeks on the diets than that in subjects who consumed the control meal, as shown in Figure 4A.

Tokutake et al. [10] reported that the disturbed insulin signaling cascade may be implicated in the pathways through which soluble Aβ induces Tau phosphorylation, and lent further support to the notion that correcting the insulin signal dysregulation in AD may offer a potential therapeutic approach.

After 12 weeks on treatment meals, the insulin levels of test subjects that ingested the WFBSK meal were significantly lower at 120 min after the meal than the insulin levels in subjects consuming the control meal (*p* < 0.05) (Figure 4B).

Test meal: wax-free black/brown rice, “Okunomurasaki”, super-hard brown rice “Niigata 129 gou” and ordinary brown rice “Koshihikari” blended in a ratio of 4:4:2 with 2.5% waxy black rice bran (WBB) “Shihou” and 0.3% rice oil added, cooked after high-pressure-treatment (HPT) by Echigo Seika, Co., Ltd. Control meal: “Koshihikari” cooked polished rice was prepared by Echigo Seika, Co., Ltd.

Although we measured HDL cholesterol, LDL cholesterol, insulin sensitivity and HbA1c, they did not show significant differences between subjects who consumed the test meal or the control meal.

## 4. Conclusions

In this study, as the texture of cooked brown rice is too hard and non-sticky to be table rice, we adopted novel processing technology to create “wax-free brown rice” (WFBR) and compared it with ordinary brown rice in terms of textural properties and functional ingredients. A human intervention test, in a randomized, parallel-group comparison study, was conducted by using the abovementioned meals as samples for 12 weeks, to be consumed by 24 subjects.

(1)Wax-free unpolished rice showed higher water absorption, which led to the improvement of eating quality and hydrophilic ORAC of black unpolished rice.(2)In addition to the wax-free processing, HPT (high-pressure treatment) and the addition of rice oil were effective at improving the eating quality of our test meal, which led to the completion of the human intervention test without dropout in terms of palatability.(3)Wax-free processing, HPT and addition of WBB (waxy black rice bran) strengthened the biofunctionality of our test meal, WFBSK.(4)After 12 weeks of human intervention testing, the postprandial blood insulin levels of the test subjects were significantly lower at 120 min than those of subjects who consumed the control diet (*p* < 0.05), which would prevent hyperinsulinemia, one of the causes of the shortage of insulin in the brain, and contribute to improvement in the cognitive domain of language memory.(5)The results of Cognitrax demonstrated that consuming WFSFK meal was significantly effective at improving the cognitive domain of “language memory” after 12 weeks (*p* < 0.05).

In conclusion, our test meal showed the possibility of controlling dementia and type 2 diabetes mellitus in the 12-week human intervention test. This result would lead to the prevention of dementia through a diet based on boiled rice. As rice is a staple food for many people in the world, this would support not only satiety but also a healthy life without dementia.

## Figures and Tables

**Figure 1 foods-11-00818-f001:**
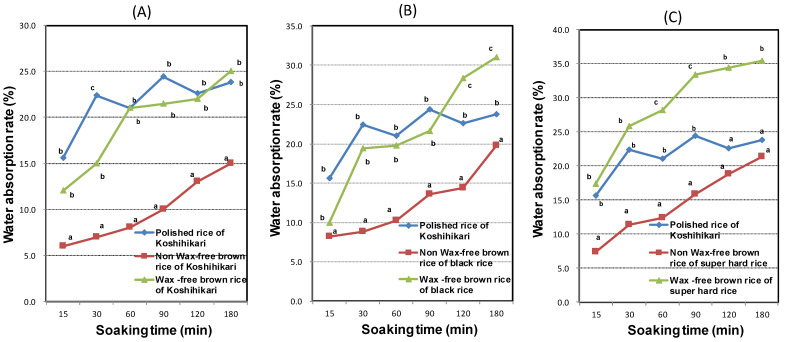
Water absorption rates of three kinds of wax-free brown rice and brown rice. Among the water absorption rates after in same soaking time, different letters (a, b, etc.) denote a statistically significant difference. Water absorption of polished Koshihikari rice is shown as a control (blue line) (**A**); water absorption rates of brown Koshihikari rice and wax-free brown Koshihikari rice (**B**); water absorption rates of brown Okunomurasaki (black) rice and wax-free brown Okunomurasaki rice (**C**); water absorption rates of brown Niigata 129 go (super hard) rice and wax-free brown Niigata 129 go rice.

**Figure 2 foods-11-00818-f002:**
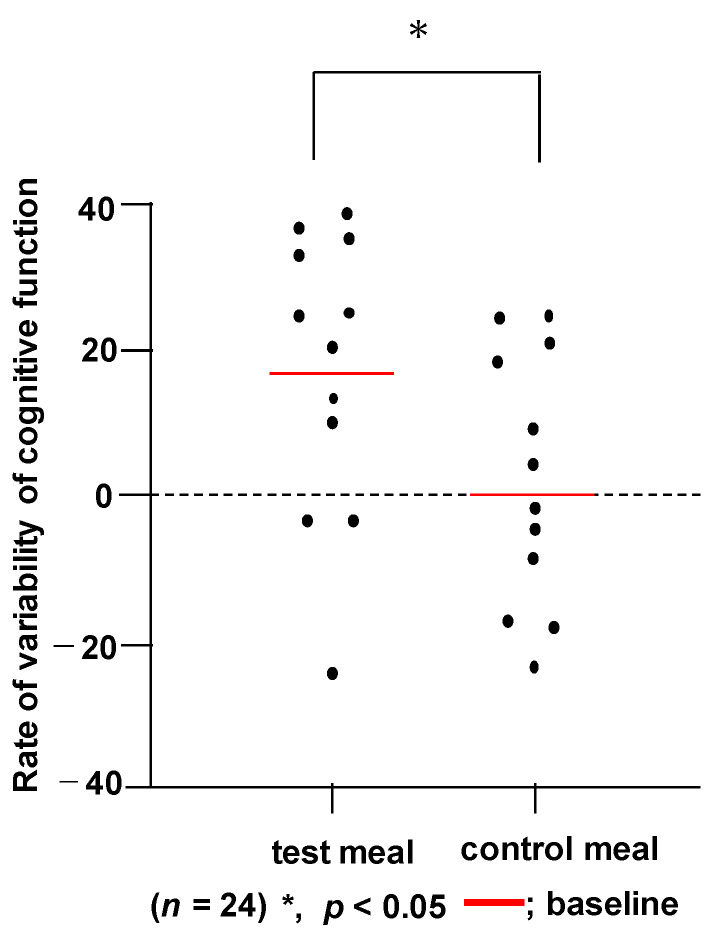
Language memory test.

**Figure 3 foods-11-00818-f003:**
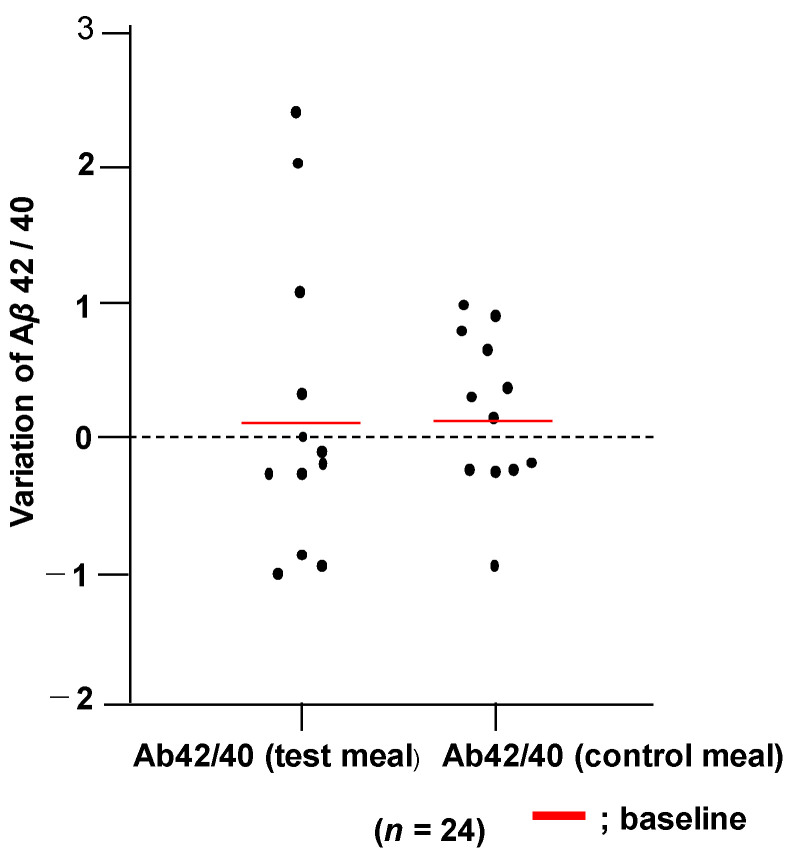
Changes of variation of amyloid β-42/4 peptide ratio by ELISA after 12 weeks.

**Figure 4 foods-11-00818-f004:**
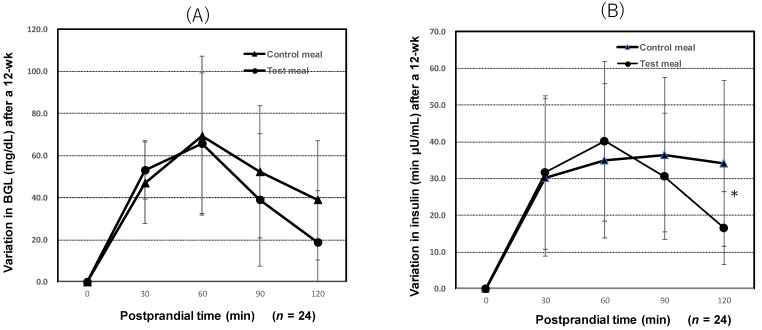
Variation in the BGL and insulin in 24 subjects after eating blending cooked rice of test meal (WFBSK) and polished rice of control after 12 weeks. (**A**) variation in postprandial BGL. (**B**) variation in postprandial insulin secretion. * means significant difference (*p* < 0.05).

**Table 1 foods-11-00818-t001:** Textural properties of various cooked brown rice and wax-free brown rice.

	Hardness	Toughness	Adhesion	Stickiness
	×10^5^ [N/cm^2^]	×10^5^ [N/cm^2^]	×10^5^ [N/cm^2^]	×10^5^ [N/cm^2^]
Cooked Koshihikari rice (BR)	0.014 ± 0.002 a	0.032 ± 0.004 a	0.020 ± 0.007 a	0.020 ± 0.003 a
Cooked Koshihikari rice (WFR)	0.009 ± 0.002 b	0.027 ± 0.004 a	0.029 ± 0.005 a	0.024 ± 0.003 b
Cooked black rice (BR)	0.021 ± 0.005 a	0.039 ± 0.005 a	0.015 ± 0.003 a	0.016 ± 0.001 a
Cooked black rice (WFR)	0.020 ± 0.005 a	0.036 ± 0.003 a	0.019 ± 0.004 a	0.022 ± 0.002 b
Cooked super-hard rice (BR)	0.047 ± 0.007 a	0.040 ± 0.004 a	0.000 ± 0.000 a	0.000 ± 0.000 a
Cooked super-hard rice (WFR)	0.025 ± 0.005 b	0.033 ± 0.008 b	0.001 ± 0.000 a	0.005 ± 0.006 b

Brown rice, BR; wax-free brown rice, WFR; within each measure (hardness, toughness, adhesion and stickiness) in the same column and each sample, different letters (a, b, etc.) denote statistically significant differences.

**Table 2 foods-11-00818-t002:** Fatty acid compositions of six kinds of japonica rice bran.

		Koshihikari	Black Rice	Waxy Black Rice	Super-Hard Rice	Red Rice	Waxy Red Rice
14:0	Myristic acid (%)	0.3 ± 0.0 a	0.4 ± 0.0 b	0.5 ± 0.0 b	0.3 ± 0.0 a	0.3 ± 0.0 a	0.3 ± 0.0 a
16:0	Palmitic acid (%)	16.7± 0.0 a	16.3 ± 0.0 a	17.6 ± 0.0 b	18.2 ± 0.0 b	17.7 ± 0.0 b	17.3 ± 0.0 b
16:1	Palmitoleic acid (%)	0.1 ± 0.0 a	0.1 ± 0.0 a	0.2 ± 0.0 a	0.2 ± 0.0 a	0.2 ± 0.0 a	0.2 ± 0.0 a
18:0	Stearic acid (%)	1.5 ± 0.0 a	2.0 ± 0.0 b	1.9 ± 0.0 b	2.0 ± 0.0 b	1.7 ± 0.0 b	1.8 ± 0.0 b
18:1	Oleic acid (%)	41.2 ± 0.1 b	46.0 ± 0.0 a	43.9 ± 0.1 a	45.3 ± 0.1 a	39.3 ± 0.0 b	40.5 ± 0.0 b
18:2n-6	Linoleic acid (%)	36.7 ± 0.1 a	32.0 ± 0.0 b	31.7 ± 0.0 b	30.6 ± 0.1 b	36.6 ± 0.0 a	36.1 ± 0.0 a
18:3n-3	α- linolenic acid (%)	1.1 ± 0.0 a	0.8 ± 0.0 b	1.3 ± 0.0 a	1.1 ± 0.0 a	1.4 ± 0.0 a	1.3 ± 0.0 a
20:0	Arachidic acid (%)	0.7 ± 0.0 a	0.8 ± 0.0 a	0.8 ± 0.0 a	0.7 ± 0.0 a	0.6 ± 0.0 a	0.7 ± 0.0 a
20:1	Icosenoic acid (%)	0.5 ± 0.0 a	0.5 ± 0.0 a	0.6 ± 0.0 a	0.5 ± 0.0 a	0.5 ± 0.0 a	0.5 ± 0.0 a
22:0	Behenic acid (%)	0.4 ± 0.0 a	0.4 ± 0.0 a	0.4 ± 0.0 a	0.3 ± 0.0 a	0.4 ± 0.0 a	0.4 ± 0.0 a
24:0	Lignoceric acid (%)	0.8 ± 0.0 a	0.7 ± 0.0 a	0.9 ± 0.0 a	0.7 ± 0.0 a	0.7 ± 0.0 a	0.7 ± 0.0 a

Within each measure (myristic acid, palmitic acid, etc.) in the same column and each cultivar, different letters (a, b, c, etc.) denote statistically significant differences.

**Table 3 foods-11-00818-t003:** The principal compositions of three kinds of cooked rice for the human test.

	A	B	C
Energy (kcal/100 g)	143.0 ± 2.8 a	148 ± 1.2 a	147 ± 1.1 a
Protein (g/100 g)	3.5 ± 0.2 a	3.1 ± 0.2 a	2.4 ± 0.1 b
Lipid (g/100 g)	1.5 ± 0.1 a	1.4 ± 0.1 a	0.5 ± 0.1 b
Carbohydrates (g/100 g)	34.0 ± 0.4 a	35.0 ± 0.2 a	33.4 ± 0.1 a
Sugar (g/100 g)	24.2 ±1.1 a	26.4 ± 1.0 a	33.0 ± 1.0 b
Insoluble dietary fiber (g/100 g)	9.5 ± 0.6 a	7.8 ± 0.2 b	0.4 ± 0.4 c
Water soluble dietary fiber (g/100 g)	0.3 ± 0.1 a	0.8 ± 0.1 b	0.1 ± 0.1 c
Moisture (g/100 g)	60.6 ± 0.5 a	59.9 ± 0.4 a	63.6 ± 0.3 a
Ash (g/100 g)	0.6 ± 0.0 a	0.6 ± 0.0 a	0.1 ± 0.0 b

A: Cooked WFBSK rice with 2.5% WBB and 0.3% rice oil added by HPT. B: Cooked BSK rice with 2.5% WBB and 0.3% rice oil added by HPT. C: Cooked polished Koshihikari rice. Waxy black rice bran, WBB; high-pressure treatment, HPT. Blend with wax-free black/brown rice, super-hard brown rice and Koshihikari brown rice (4:4:2), WFBSK. Blend with black/brown rice, super-hard brown rice and Koshihikari brown rice (4:4:2), BSK. Within each measure (energy, protein, etc.) in the same column, different letters (a, b, c, etc.) denote statistically significant differences. Dietary fiber: enzymatic-gravimetric method (Prosky variant). Protein: Kjeldahi method. Lipid: Gas chromatography. Ash: Inductively coupled plasma atomic emission spectrometry. Moisture: Drying method by heating.

**Table 4 foods-11-00818-t004:** Biofunctional properties of three kinds of cooked rice for human test.

	H-ORAC	L-ORAC	Total ORAC	Anthocyanin	Polyphenol	RS	Free	β-Secretase
	μmol	μmol	μmol		GAEmg/		Ferulic Acid	Inhibition Rate (%)
	TE/100 gFW	TE/100 gFW	TE/100 gFW	(mg/g)	100 gFW	(%)	(mg/100 g)	(0.27μg-eq/µL)
A	77.5 ± 3.4 b	5.2 ± 0.4 b	82.7 ± 3.8 b	8.03 ± 0.03 b	8.30 ± 0.20 b	5.1 ± 0.1 b	0.60 ± 0.00 b	9.2 ± 0.3 b
B	78.0 ± 1.3 b	4.9 ±0.0 b	82.9 ± 1.3 b	5.84 ± 0.02 b	7.87 ± 0.20 b	5.1 ± 0.1 b	0.57 ± 0.01 b	8.8 ± 0.3 b
C	0.0 ± 0.0 a	1.1 ± 0.1 a	1.1 ± 0.2 a	0.00 ± 0.00 a	0.00 ± 0.00 a	0.7 ± 0.0 a	0.30 ± 0.00 a	0.0 ± 0.0 a

A: Cooked WFBSK rice with 2.5% WBB and 0.3% rice oil added by HPT. B: Cooked BSK rice with 2.5% WBB and 0.3% rice oil added by HPT. C: Cooked polished Koshihikari rice. Waxy black rice bran, WBB; high-pressure treatment, HPT. Blend with wax-free black/brown rice, super-hard brown rice and Koshihikari brown rice (4:4:2), WFBSK. Blend with black/brown rice, super-hard brown rice and Koshihikari brown rice (4:4:2), BSK. Within each measure (H-ORAC, L-ORAC, etc.) in the same column, different letters (a, b, c, etc.) denote statistically significant differences.

**Table 5 foods-11-00818-t005:** Textural properties of three kinds of cooked rice for human test.

	Hardness	Toughness	Adhesion	Stickiness
	×10^5^ [N/cm^2^]	×10^5^ [N/cm^2^]	×10^5^ [N/cm^2^]	×10^5^ [N/cm^2^]
A	0.002 ± 0.00 a	0.025 ± 0.00 b	0.007 ± 0.00 a	0.010 ± 0.00 b
B	0.002 ± 0.00 a	0.024 ± 0.01 b	0.008 ± 0.00 a	0.012 ± 0.01 b
C	0.002 ± 0.00 a	0.018 ± 0.00 a	0.007 ± 0.00 a	0.006 ± 0.00 a

A: Cooked WFBSK rice with 2.5% WBB and 0.3% rice oil added by HPT. B: Cooked BSK rice with 2.5% WBB and 0.3% rice oil added by HPT. C: Cooked polished Koshihikari rice. Waxy black rice bran, WBB; high-pressure treatment, HPT. Blend with wax-free black/brown rice, super-hard brown rice and Koshihikari brown rice (4:4:2), WFBSK. Blend with black/brown rice, super-hard brown rice and Koshihikari brown rice (4:4:2), BSK. Within each measure (hardness, toughness, etc.) in the same column, different letters (a, b, c, etc.) denote statistically significant differences.

**Table 6 foods-11-00818-t006:** Result of sensory test of three kinds of the blended meals.

	Appearance	Aroma	Hardness	Taste	Stickiness	Overall Evaluation
A	3.13 ± 0.85 a	3.13 ± 0.25 a	2.38 ± 0.48 a	3.85 ± 0.02 a	3.35 ± 0.02 a	3.85 ± 0.01 a
B	3.25 ± 0.29 a	3.13 ± 0.25 a	2.88 ± 0.48 a	3.30 ± 0.01 b	3.25 ± 0.01 a	3.31 ± 0.02 b
C	3.00 ± 0.00 a	3.00 ± 0.00 a	3.00 ± 0.00 a	3.00 ± 0.00 b	3.00 ± 0.00 a	3.00 ± 0.00 b

A: Cooked WFBSK rice with 2.5% WBB and 0.3% rice oil added by HPT. B: Cooked BSK rice with 2.5% WBB and 0.3% rice oil added by HPT. C: Cooked WFBSK rice with 2.5% WBB added by HPT. Waxy black rice bran, WBB; high-pressure treatment, HPT. Blend with wax-free black/brown rice, super-hard brown rice and Koshihikari brown rice (4:4:2), WFBSK. Blend with black/brown rice, super-hard brown rice and Koshihikari brown rice (4:4:2), BSK. Within each measure (appearance, aroma, etc.) in the same column, different letters (a, b, c, etc.) denote statistically significant differences. a, b < 0.05, *n* = 10.

**Table 7 foods-11-00818-t007:** Comparison between test and control meals for human intervention test items.

	Test Meal	SD	Control Meal	SD	Paired-*t*
	(*n* = 12)		(*n* = 12)		*p*-Value
Age	60.1	6.3	58.0	4.8	0.4345
BMI (kg/m^2^)	21.9	2.4	21.0	2.3	0.6763
FBG (mg/mL)	94.9	5.2	95.2	9.3	0.9306
HbA1c	5.6	0.2	5.6	0.3	0.7525
MMSE	29.0	1.0	29.3	0.8	0.4649

BMI, body mass index; FBG, fasting blood glucose; HbA1c, hemoglobin A1c; MMSE, mini-mental state examination.

**Table 8 foods-11-00818-t008:** Results of Cognitrax.

	Test Meal	Control Meal
Comprehensive memory	7.25 ± 13.87 a	−0.83 ± 12.19 a
Language memory	16.00 ± 18.35 a	1.00 ± 15.08 b
Visual memory	−3.67 ± 12.28 a	−2.75 ± 14.05 a
Cognitive function speed	4.08 ± 7.18 a	0.17 ± 7.33 a
Reaction time	1.83 ± 7.90 a	7.33 ± 8.90 a
Comprehensive attention	6.42 ± 11.98 a	1.17 ± 17.83 a
Cognitive flexibility	7.17 ± 13.07 a	4.50 ± 17.48 a
Processing speed	5.50 ± 12.90 a	5.33 ± 5.77 a
Execution mechanism	7.33 ± 12.76 a	4.75 ± 15.71 a
Simple attention	2.75 ± 13.80 a	−8.50 ± 41.58 a
Movement speed	1.25 ± 6.54 a	−2.92 ± 7.91 a
Finger tapping (left hand)	0.50 ± 4.06 a	−6.00 ± 9.72 b
Symbol digit coding	−7.50 ± 18.30 a	2.83 ± 8.39 a
Stroop test	−6.70 ± 27.2 a	11.42 ± 18.58 a

Within each measure (comprehensive memory, language memory, etc.) in the same column, different letters (a, b, etc.) denote statistically significant differences. a, b < 0.05.

## Data Availability

The datasets generated for this study are available on request to the corresponding author.

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
