# Peer review of "Possibility for Prevention of Type 2 Diabetes Mellitus and Dementia Using Three Kinds of Brown Rice Blends after High-Pressure Treatment"

_foods, 2022, doi:10.3390/foods11060818_

Round 1

Reviewer 1 Report

Authors need to clarify and ask themselves again what question is this paper going to answer, and only do the experiment can/may address this question. Try to many things at the same time can cause mess.  Authors list all data, but did not give detailed explanations and comparisons. The benefits of black rice and hard rice are not outstanding and convincing. Spelling and grammar also need to be checked. 

In Figure 1, the spelling errors are ‘super kard rice’, ‘Wax-free brawn rice’.  Using names called ‘Brown rice’ and ‘Wax-free brown rice’ is confusing, recommend to use ‘Non Wax-free brown rice’ and ‘Wax-free brown rice’. 

Line 273-284, authors should explain/propose the reasons of those differences rather than just giving the data. 

Line 294, to be accurate, the title should be ‘Textural properties’ rather than ‘Physical properties’.  

Line 195, authors need to state the definitions of Hardness and Roughness, Adhesion and Stickiness, how those four parameters are measured. 

Line 301, where is rice color data? It can be put in the Supplementary File. 

Line 300 and 305, authors use ‘improve physical properties’. Does that mean increase stickiness and reduce hardness? Use more accurate and scientific words. Like using ‘every day’ is also too subjective. 

Line 314, authors listed the fatty acid compositions. But it is lack of the comparisons between samples, and highlights of healthy effects.  Not all acids are healthy.  For example, black rice has high concentration of ***, thus may have *** benefits. 

Line 338, authors measured the compositions of three rice blends rather than cooked rice. Thus the tile is wrong, it should be ‘components of 3 kinds of rice blends’. 

Line 361, using ‘ingredients’ is not proper. Use ‘compositions’. 

Author Response

Answer to the reviewer 1

Authors need to clarify and ask themselves again what question is this paper going to answer, and only do the experiment can/may address this question. Try to many things at the same time can cause mess.  Authors list all data, but did not give detailed explanations and comparisons. The benefits of black rice and hard rice are not outstanding and convincing. Spelling and grammar also need to be checked. 

Thank you for your valuable comments. As we revised our manuscript according to the comments, we appreciate deeply if you kindly review the revised manuscript.

In Figure 1, the spelling errors are ‘super kard rice’, ‘Wax-free brawn rice’.  Using names called ‘Brown rice’ and ‘Wax-free brown rice’ is confusing, recommend to use ‘Non Wax-free brown rice’ and ‘Wax-free brown rice’. 

Thank you for your valuable advice. We corrected typo and revised the legend according to your comment.

Line 273-284, authors should explain/propose the reasons of those differences rather than just giving the data. 

We added the explanation in L346 to 347.

Line 294, to be accurate, the title should be ‘Textural properties’ rather than ‘Physical properties’.  

Thank you for your advice. We revised in L361,

Line 195, authors need to state the definitions of Hardness and Toughness, Adhesion and Stickiness, how those four parameters are measured. 

We added definition on Hardness, Toughness, Adhesion, and Stickiness in L.362 to 363. These 4 parameters are often used in the measurement of cooked rice grains in Japan because Asian people prefer soft and sticky grains as boiled rice.

Line 301, where is rice color data? It can be put in the Supplementary File. 

We added color data in the Supplemental Table 1.

Line 300 and 305, authors use ‘improve physical properties’. Does that mean increase stickiness and reduce hardness? Use more accurate and scientific words. Like using ‘every day’ is also too subjective. 

 Yes. As we explained before, North-east Asian consumers eat boild rice almost every day and they prefer soft and sticky rice. We added “Asian” and ”almost” to express more scientifically.

Line 314, authors listed the fatty acid compositions. But it is lack of the comparisons between samples, and highlights of healthy effects.  Not all acids are healthy.  For example, black rice has high concentration of ***, thus may have *** benefits. 

Thank you for your advice. We added new reference and compared and evaluated rice samples in terms of bio-functionality from L406 to L412.

Line 338, authors measured the compositions of three rice blends rather than cooked rice. Thus the title is wrong, it should be ‘components of 3 kinds of rice blends’. 

Thank you. We corrected the title in L422.

Line 361, using ‘ingredients’ is not proper. Use ‘compositions’.

Thank you. We corrected the title of the Table 3 in L442.

Reviewer 2 Report

The submission will benefit immensely from a professional English grammar revision. Reviewer has made just a few examples, as drawn off the first page. There are far too many unclear phrases and grammar or tense issues to accept the paper in its current form. This could be a problem of software converting Japanese to English?

Line 16. Please consider … we investigated the potential to prevent (decrease?) diabetes and demetia by consumption of the ___ [give specific treatment product name (not “the rice product”)]_____. (instead of “… investigated to prevent diabetes and dementia by the rice product.”) 

M&M. 

Excessive use of acronyms and an acronym matrix might help the reader (Yes, I know they are in the “Abbreviations” but it is still difficult). Also, some clarification of M&M (Section 2.2) with regard to acronyms (e.g. 10% removal: well, is this white rice (WR) or wax-free black rice, WFBR)? 

2.2. Please indicate if 10% bran removal in all varieties is indeed white polished rice (WR). Furthermore, it would follow convention to call these WR throughout the paper (as opposed to WFR, which is a little confusing). 

Also fix terminology because using brown and black and rice twice is cumbersome. For example, “The dehusked waxy black rice (Shihou …. 

Please be consistent in nomenclature. For example ‘Niigata129go’ versus ‘Niigata 129 go’. 

Shihou is mentioned in the M&M, then only twice throughout the paper. This seems to be an important variety and treatment, which, I believe then gets re-termed into "WFBR"?

Most literature in the US uses variety as opposed to cultivar but these are often interchanged. 

Figure 1. The in-text legend descriptions indicating the 3 varieties are not the same on the imbedded legends, per panel. Variety names in each panel needs to be fixed (they are not allKoshihikari). All occurrences of “brawn” need to be “brown”. Must be fixed. 

2.10 and 2.11 and 2.13 are very scantily depicted to the reader. Please at least provide a citation. 

Several typos. 

Some R&D appears copy-paste-like from another document, and text does not flow smoothly with one coherent train of thought.

Strange how M&M-like information is found immediately above the Conclusion… 

Many Tables are in all bold font, and some seem too wide. Can this be fixed?

Author Response

Answer to the reviewer 2

We are grateful for your kind and valuable comments. As we revised our manuscript according to the comments, we appreciate if you kindly review it again.

The submission will benefit immensely from a professional English grammar revision. Reviewer has made just a few examples, as drawn off the first page. There are far too many unclear phrases and grammar or tense issues to accept the paper in its current form. This could be a problem of software converting Japanese to English?

We checked and corrected the typing and grammatical problems.

Line 16. Please consider … we investigated the potential to prevent (decrease?) diabetes and dementia by consumption of the ___ [give specific treatment product name (not “the rice product”)]_____. (instead of “… investigated to prevent diabetes and dementia by the rice product.”) 

We revised our manuscript in L.15-16 to “we investigated the potential to prevent type 2 diabetes and dementia using bio-functional boiled rice” according to the comment.

M&M. 

Excessive use of acronyms and an acronym matrix might help the reader (Yes, I know they are in the “Abbreviations” but it is still difficult). Also, some clarification of M&M (Section 2.2) with regard to acronyms (e.g. 10% removal: well, is this white rice (WR) or wax-free black rice, WFBR)? 

We decreased excess use of acronyms in M&M. For example, we changed “WFBSKB rice (WFBR : SHBR : KBR = 4 : 4 : 2) was added with 2.5 % (w/w)” to “WFBSKB rice (wax-free black rice (WFBR) : un-polished super-hard rice (SHBR) : unpolished Koshihikari rice (KBR) = 4 : 4 : 2) was added with 2.5 % (w/w)”. And we prepared black rice bran in 2.2. in order to use on rice cooking to improve eating quality and anti-oxidation capacity. 

2.2. Please indicate if 10% bran removal in all varieties is indeed white polished rice (WR). Furthermore, it would follow convention to call these WR throughout the paper (as opposed to WFR, which is a little confusing). 

As we described above, we prepared black rice bran in 2.2. in order to use on rice cooking by adding the bran to improve eating quality and anti-oxidation capacity. In the case of black rice, rice grain is slightly purple after polishing to the milling yield of 90%. In this research we used wax-free brown rice for black rice (Wax-free brown rice: milling yield is more than 99%). We are sorry for causing confusion using the word “wax-free brown rice” and white rice. We changed to use the word “wax-free unpolished rice” in the place of “wax-free brown rice.”

Also fix terminology because using brown and black and rice twice is cumbersome. For example, “The dehusked waxy black rice (Shihou …. 

Thank you for your kind proposal. According to the comments by the other reviewer, we replaced “brown rice” with ”unpolished rice” to avoid the confusion.

Please be consistent in nomenclature. For example, ‘Niigata129go’ versus ‘Niigata 129 go’. 

Thank you for your comment. We unified to use “Niigata 129 go”.

Shihou is mentioned in the M&M, then only twice throughout the paper. This seems to be an important variety and treatment, which, I believe then gets re-termed into "WFBR"?

We used “Okunomurasaki” (non-waxy black rice) for WFBR (wax-free brown rice) and used “Shihou” (waxy black rice) only for getting rice bran because its fatty acid composition is more suitable as shown in Table 2.

Most literature in the US uses variety as opposed to cultivar but these are often interchanged. 

According to the comment, we added Japanese registration number of each rice cultivar.

Figure 1. The in-text legend descriptions indicating the 3 varieties are not the same on the imbedded legends, per panel. Variety names in each panel needs to be fixed (they are not all Koshihikari). All occurrences of “brawn” need to be “brown”. Must be fixed. 

We are sorry for typo and corrected them. According to the comment by the other reviewer, we revised the expression of imbedded legends. A: Koshihikari, B; Black rice C; Super hard rice

2.10 and 2.11 and 2.13 are very scantily depicted to the reader. Please at least provide a citation. 

Thank you for your comment. We revised and added explanation in 2.10, 2.11, and 2.13.

Several typos. 

We are sorry and corrected the typo.

Some R&D appears copy-paste-like from another document, and text does not flow smoothly with one coherent train of thought.

We are sorry. We revised R&D.

Strange how M&M-like information is found immediately above the Conclusion… 

We deleted most of M&M information and revised.

Many Tables are in all bold font, and some seem too wide. Can this be fixed?

We are sorry. We corrected them.

Reviewer 3 Report

My comments are in the attached file.

Author Response

The methods and results are not well-presented and need to be improved. I found the overall flow of the paper very disorganized.

According to the comments, we revised the methods and results. We tried to clarify the overall flow of the manuscript.

1)  Title:

  1. instead of “by the use of” it is better to use “using”.

We replaced “by the use of” with the expression of ”using”.

  1. The title is vague: I suggest reformatting the title to be more specific and to avoid misinformation; using these rice cultivars to prevent diabetes and dementia or to control them?

Thank you for your suggestion. We changed the title to “Possibility for prevention of type two diabetes mellitus and dementia using three kinds of brown rice blends after high pressure treatment.”

  1. Please specify what type of diabetes!

We used the expression “type two diabetes mellitus” in the title.

2)  Abstract

  1. Abstract clears out my comments on the title but as I said, title should not be mis-leading.

We changed the title as above-mentioned, and we revised the abstract.

  1. It is not clear when the “single-dose test for postprandial blood glucose and insulin secretion” was conducted during the Please clarify that!

We added “at the end of human intervention test.” in the abstract.

  1. Please add one sentence of conclusion ad the end of the abstract!

We added “In conclusion, our test meal showed possibility for preventing dimentia and type two diabetes mellitus by the 12-week human intervention test” at the end of conclusion.

3)  Introduction:

  1. Intro is too long and the missing the introducing the intervention.

We revised the introduction and added the explanation of intervention test.

  1. For ref 1-4, please use more recent references

We added 4 more recent references.

  1. Line 50-51: Reference needed

We added references.

  1. Line 58-59: “of which components are minerals, polyphenols, flavonoids, vitamins, omega-3 PUFAs, ”,

should be revised because food composition is mor than just what is mentioned here.

We revised.

  1. Line 62: please mention the scientific name of rice (Oryza sativa ) in front of it

We added scientific name.

  1. I would re-order the introduction flow in a way that it reads more fluent:

一 i. merge the paragraph about Dementia and Cognitive function together!

一 ii. Then talk about the GI and food with low GI

一 iii. Then talk about rice and rice bran and how it is beneficial for T2Dand AD control and prevention etc. in one paragraph – right now there are four different paragraphs about rice and not in order.

      We revised according to the comment.

  1. Line 76-78: please update the rice utilization + add reference!

We added new utilization of low-GI rice products (noro, Akaishi, Nakamura?)

  1. Line 90: reference needed

We added reference; J Nutr. 2006 Aug;136(8):2220-5. doi: 10.1093/jn/136.8.2220.

An anthocyanin-rich extract from black rice enhances atherosclerotic plaque stabilization in apolipoprotein E-deficient mice

Xiaodong Xia 1, Wenhua Ling, Jing Ma, Min Xia, Mengjun Hou, Qing Wang, Huilian Zhu, Zhihong Tang

Affiliations expand

  1. Line 107: “rice product” è “rice products”.

We changed “rice product” to ”rice products”.

  1. Lines 107-110: please reformat the objective and hypothesis of the It is vague, as it is now.

We added objective and hypothesis of this study.

4)  Material and methods

  1. All the measurement methods should be described briefly. Referring them with another publication is not For example, Section 2.7. Briefly mention what the Folin–Ciocalteu method is! and so on, on other sections. A good example is Section 2.14.

Thank you for your advice. We added explanations for other experiments.

  1. b) Section 2.1. It would be good to see the characteristics of these rice varieties, particularly their genetic

accession numbers, as a supplementary table for the reproducibility purposed. For example, something like

table 5 of this publication https://pubmed.ncbi.nlm.nih.gov/30304872/.

       We added the registration number in Japan for three rice cultivars from L135 to L138.

  1. c) Section 2.3. the first sentence does not read well, grammatically. Please fix it!

       Thank you for your comment. We revised as follows;

        We prepared cooked rice by blending wax-free black rice (WFBR), super-hard brown rice (SHBR) and Koshihikari (high-quality rice) brown rice (KBR) (4:4:2) (WFBSKB) WFBSKB rice (WFBR : SHBR : KBR = 4 : 4 : 2) was added with 2.5 % (w/w) waxy black rice bran (WBB) and 0.3 % (w/w) rice oil (Tsuno Food Industrial Co., Ltd, Wakayama, Japan) , which was obtained by treating at 200 MPa for 2 min in a high pressure machine (Ishikawajima-Harima Heavy Industries Co., Ltd., Tokyo, Japan), and cooked rice was prepared by Echigo Seika, Co., Ltd.

  1. d) Section 2.6. Spell pout the RS in the heading!

       We revised to “Measurement of RS (resistant starch) of cooked rice” in L.191.

  1. e) Section 2.8. The abbreviations should go at the end i.e., “Measurement of hydrophilic and lipophilic (H-ORAC

and L-ORAC) oxygen radical absorbance capacity” →“Measurement of hydrophilic and lipophilic oxygen

radical absorbance capacity (H-ORAC and L-ORAC)”.

We revised to “Measurement of hydrophilic and lipophilic (H-ORAC and L-ORAC) oxygen radical

 absorbance capacity (H-ORAC and L-ORAC)” in L216,217.

  1. f) Section 2.9. what do you mean by “Physical properties”? Elaborate here please! I had to go the result section

to know what you meant.

       We revised to “Textural properties.”

  1. g) Section 2.13. Spelling/typo error. Fix it please! Please fit all the typos throughout the manuscript. I cannot

spot them all.

      We revised “fiber” to “fibre” in L84, L250, and L251.

  1. h) Section 2.16 is part of the trial and should come after the study design section.

     We revised according to the comment.

  1. i) Section 2.17. Please add the diagram of study design or a CONSORT flow.

    We added diagram in supplemental Figure 2.

  1. j) Section 2.17. please elaborate the study food in terms of type of food, amount, etc. For example, x number of

meals were developed for each of the two study arms (control and intervention). The control meals contained

the same ingredients as the intervention meals, but did not have x and y ingredients included. Consumption of one meal was equal to the daily required intervention amount across control, and the intervention (X grams). Control and intervention recipes were designed to provide comparable quantities of calories and

macronutrients (REFERENCE needed). Complete nutritional analysis was done for each recipe using the XXXX

program (e.g. Nutritionist Pro).

We explained about the meal in L.292 to L295 and L545.  We added the components of each meal in the supplemental Table 1. We described it in L.294 and L295 because we tried to use the same amount of carbohydrate for each meal. 

  1. k) Section 2.17. how many times did the subjects consume the “test food or control food” throughout of the

study span? The food development and how the authors did that is missing too.

All the subjects consumed 84 packs (1(pack of boiled rice) X 7(days)X12(weeks) ) throughout the study.

We developed the test meal based on our method (reference 45, 62 ).

  1. l) Section 2.17. line 238: “ate”→“consumed”. Instead of using the verb “to eat”, please use “to consume”

throughout the manuscript.

We revised in L293.

  1. m) Section 2.17. supplementary Table 1 should be mentioned here in this section!

       We mentioned in L292 and L294, L295.

5) Results and Discussions

I don’t have time to go through the whole results and discussion. I found the presentation of the result

disorganized and hard to follow the order of the results based on the sequence was presented in the Method

section. Some of the results were missing: for example I was not able to find any results related to the some of the blood measurements at were brought up in the methods. I think the discussion aspect needs still work

We revised Materials & Methods according to the comment, which made it easy to check the relation between M&M and R&D. And we added the results on HDL cholesterol, LDL cholesterol, insulin sensitivity and HbA1c

In L622 to 624.

  1. and it does not come out why measuring that specific property in the tested rice are important!

De n

As we reported in our previous paper (reference 45, 62), we measured textural property and sensory property in terms of eating quality, we measured anti-oxidant capacity, contents of resistant starch and dietary fibre in terms of bio-functionality of rice.

  1. Section 3.1. Please no just abbreviation in the heading , throughout the manuscript.

We revised in L338.

  1. Why are the text of Tables 1 and 2 bold?

We are sorry. We revised.

6) Conclusion:

  1. I liked the bullet point conclusion style but they are not conclusion but the summary of the results. please write a

We revised our conclusion according to the comment.

  1. proper conclusion and how these findings are important and applicable I terms of human health.

We revised our conclusion according to the comment and we described the meaning of our results in L622 to L624.

Round 2

Reviewer 3 Report

Thanks for improving the manuscript.  It looks good to me now. However, just a minor comment that it would be nice if you use a bit more scientific language to describe the importance of this study. I found it very much layman language as it is now.

Author Response

We are grateful for your valuable comments. According to your comments, we revised conclusion by using scientific words and expressions.